# Pilot Behavior Recognition Based on Multi-Modality Fusion Technology Using Physiological Characteristics

**DOI:** 10.3390/bios12060404

**Published:** 2022-06-12

**Authors:** Yuhan Li, Ke Li, Shaofan Wang, Xiaodan Chen, Dongsheng Wen

**Affiliations:** National key Laboratory of Human Machine and Environment Engineering, School of Aeronautical Science and Engineering, Beihang University, Beijing 100191, China; ffflora@buaa.edu.cn (Y.L.); by2005516@buaa.edu.cn (S.W.); xiaodan_chen@buaa.edu.cn (X.C.)

**Keywords:** MTF, physiological, behavior recognition, pilot, machine learning, multi-modal

## Abstract

With the development of the autopilot system, the main task of a pilot has changed from controlling the aircraft to supervising the autopilot system and making critical decisions. Therefore, the human–machine interaction system needs to be improved accordingly. A key step to improving the human–machine interaction system is to improve its understanding of the pilots’ status, including fatigue, stress, workload, etc. Monitoring pilots’ status can effectively prevent human error and achieve optimal human–machine collaboration. As such, there is a need to recognize pilots’ status and predict the behaviors responsible for changes of state. For this purpose, in this study, 14 Air Force cadets fly in an F-35 Lightning II Joint Strike Fighter simulator through a series of maneuvers involving takeoff, level flight, turn and hover, roll, somersault, and stall. Electro cardio (ECG), myoelectricity (EMG), galvanic skin response (GSR), respiration (RESP), and skin temperature (SKT) measurements are derived through wearable physiological data collection devices. Physiological indicators influenced by the pilot’s behavioral status are objectively analyzed. Multi-modality fusion technology (MTF) is adopted to fuse these data in the feature layer. Additionally, four classifiers are integrated to identify pilots’ behaviors in the strategy layer. The results indicate that MTF can help to recognize pilot behavior in a more comprehensive and precise way.

## 1. Introduction

With the continuous development of autopilot systems and flight assistance systems, pilot performance and air safety have been improved significantly. For example, the DLR project ALLFlight provides pilot assistance during all phases of flight [1] and includes an automatic aircraft upset-recovery system (AURS) that supports the pilot in recovering from any upset in a manner that is both manually assisted and automatic [2] and an assisting flight control system (FCS) that can simplify piloting, reduce pilot workload, and improve the system’s reliability [3]. Owing to these developments, the main task of the pilot has changed from controlling the aircraft to supervising the autopilot system and making critical decisions [4]. Therefore, it is essential to improve autopilot systems via better human–machine interaction. In addition to optimizing the display interface and improving the automation technology, ensuring that the flight assistance system has a better understanding of the pilots’ physiological status is a key step to improving human–machine interaction. To correctly evaluate the effects of different flight behaviors at different difficulty levels on pilot status, it is necessary to monitor various physiological parameters of pilots under different flight behaviors and predict their workload.

Traditionally, the pilots’ workload under different flight operations is assessed based on expert interviews and professional questionnaires, such as subjective rating scales [5]. However, these indirect analyses have many problems. For example, assessing workload through questionnaire evaluation is subjective, and the result is greatly influenced by different individuals. Furthermore, the pilot sometimes needs to interrupt the flight in order to answer the questionnaire, which cannot be carried out in real scenarios. Moreover, the questionnaire can only be conducted at discrete time points and cannot provide information on continuous, task-related workload or physiological state changes [6]. To solve these problems, a more objective and time-continuous method, which recognizes the human state according to physiological parameters, has been proposed. For instance, Law [7] assessed pilot workload by analyzing electro cardio (ECG) and electroencephalogram (EEG) data and showed that RR interval (RRI) and the root mean square of successive differences (RMSSD) decreased with increasing flight mission difficulty. Feleke [8] used myoelectricity (EMG) to detect the driver’s intention to make an emergency turn and found that the driver’s EMG rose sharply before the turn. Some studies adopted a single physiological signal for the identification of human states. For example, Rahul [9] used ECG to estimate pilot fatigue, and Matthew [10] used functional near-infrared spectroscopy to identify the mental load of helicopter pilots. However, recognition by a single physiological signal suffers from poor stability, reduced data information, low reliability, and low discriminatory ability. Multiple physiological signals have therefore been proposed. For example, Pamela et al. [11] used electrodermal and electrocardiographic information to monitor the sympathetic response of drivers. Nevertheless, the research on multi-modal physiological signals for human state recognition still offers great development opportunities. Despite some studies [12,13,14] involving a combination of physiological signals, there is still no consensus on many elements, for instance, which indicators should be used as input, at which level to fuse, what kind of the classification models should be adopted [15,16], and how to deal with the problem of insufficient available data. Consequently, it is essential to create improved frameworks that provide greater robustness in identifying the status of pilots during different operations.

Besides fusing physiological information, the fusion of classification models to achieve optimal results is also a way to improve the framework. This requires the investigation of several classifiers which are to be integrated. These candidate classifiers have been used frequently in previous studies. For example, Wang [17] analyzed skin conductance, oximetry pulse, and respiration signals using Hilbert transform and a random forest classifier algorithm and evaluated the model by Accurate Rate, MSE, ROC, F1_score, Precision, and Recall. Hu [18] determined whether a driver is fatigued based on EEG signals using a gradient-boosting decision tree (GBDT), reaching 94% accuracy, and the k-nearest neighbor algorithm, support vector machine, and a neural network were also employed as a comparison. Vargaslopez [19] compared different machine learning algorithms, such as SVM and MLP, and showed that SVM obtained the best result in detecting stress after normalization.

This study has two objectives: (i) to develop a model that recognizes pilots’ behavior based on multi-modal fusion technology (MFT) that uses physiological characteristics; and (ii) to analyze the physiological indicators influenced by the pilot’s behavioral status and their association with flight difficulty. Firstly, a physiological database of pilots in different behavioral states is formed through experimental protocol design and data collection. Then, the correlation between pilot behavioral states and physiological data is analyzed in terms of raw data, time-domain features, and frequency-domain features, and the difficulty of flight behavior is evaluated. Additionally, the classification model for recognizing pilots’ behavior is designed based on MFT. Finally, the performance of each machine learning classification model and the proposed model is analyzed and compared from the perspectives of accuracy, precision, F1 score, and mean squared error (MSE). As a result, this paper makes the following contributions: (i) a new physiological data set from 14 pilots with wearable physiological measurement devices for multi-modal data input; (ii) a classification model based on MFT, the accuracy of which reaches 98.15%; and (iii) an analysis of the effects of flying behaviors on a pilot’s physiological measurements from an objective point of view.

## 2. Methods

The study was conducted according to the process of data mining [20], which is shown in Figure 1.

### 2.1. Data Construction

#### 2.1.1. Participants

In this study, data were collected from 14 Air Force flight cadets (age 21±2 years, weight 70±6 kg, all male). All subjects were professionally trained and proficient in flying the F-35 Lightning II Joint Strike Fighter used for flight testing. All subjects were right-handed, had normal vision, and had no history of cardiac, neurological, or psychiatric disease. All subjects rested well and were healthy during the experiment.

#### 2.1.2. Flight Platform and Task Details

In this study, the pilots performed various flight maneuvers in a flight simulator. The flight simulator consisted of the obutto ergonomic workstation, Saitek Pro Flight X-56 Rhino Stick (for controlling direction, roll, and pitch in the air), Saitek Pro Flight X-56 Rhino Throttle (for controlling throttle), foot pedals (for controlling direction on the ground), and flight simulation software Xplane (as shown in Figure 2). It provided a flight simulation environment with high fidelity and good immersion. The simulation scenario was 10 nautical miles from Beijing Capital Airport.

Before the experiment began, each subject had 5 min to familiarize themselves with the flight simulator. Then, the experiment started officially, and each subject flew as instructed, completing five types of operation in each flight: takeoff, level flight, hover and turn, roll, and somersault, and there was a chance that a stall scenario might occur during the operation. Each subject performed the experiment once. All the maneuvers were carried out for about 4 min to achieve a balanced data set, except for the stall scenes, which occurred randomly and took about 3 min. Operations that met the criteria were counted. After the experiment, all subjects were asked to rate the difficulty of the different flight behaviors (1 to 10, with 10 being the most difficult) to obtain their subjective opinions.

#### 2.1.3. Data Acquisition

The ErgoLAB Human Factors Physiological Recorder was used to collect physiological data from the subjects. ECG data were collected from the subject’s chest via Ag/AgCl electrodes (sampling rate of 512 Hz) and also from the earlobe via a pulse sensor (PPG) (sampling rate of 64 Hz). GSR data were collected from the palm of the subject’s right hand via Ag/AgCl electrodes (sampling rate of 64 Hz); EMG data were collected from the radial carpal extensor muscle of the subject’s right lower arm via Ag/AgCl electrodes (sampling rate of 1024 Hz); RESP data were collected from the subject’s chest cavity via a chest strap respiratory sensor (sampling rate of 64 Hz); SKT data were collected from the subject’s right lower arm via a skin temperature sensor (sampling rate of 32 Hz). All data were transmitted via Bluetooth sensors to the synchronization platform for processing.

To reduce industrial frequency interference and environmental interference, high viscosity electrodes were used to ensure good contact when collecting data. To avoid the effect of temperature and humidity on data acquisition, the laboratory was kept in a dry condition, and the temperature was maintained at 22–24 degrees Celsius. To reduce motion disturbance, the subjects were told to avoid large movements as much as possible.

### 2.2. Data Pre-Processing

Data pre-processing is a time-consuming but essential step and can help improve the classification accuracy and performance [21], as there are obvious problems in raw data, including: missing data, data noise, data redundancy, unbalanced data sets, etc. Data pre-processing methods can be summarized into three categories [22], as shown in Table 1. It is believed that the pre-processing method should be selected and adjusted according to the data set and the specific task [23,24]. Our data set was subject to time continuity, individual variability, and a high sampling rate, and was affected by motion and noise. As a result, the pre-processing methods described below were adopted.

#### 2.2.1. Normalization

Due to individual differences, each subject’s measured physiological signal exhibited a different scale or magnitude. Therefore, it was necessary to normalize the physiological signal for each subject to obtain the same scale or magnitude [25]. Normalization methods included min–max, z-score, baseline, etc. In this study, the changes in physiological indicators varied greatly between individuals during flight and at rest. Adopting the baseline method may have made the data subject to data scale inconsistency. To ensure that the data distribution was not altered and to eliminate the influence of dimensionality between indicators, the ECG signal, EDA signal GSR signal, EMG signal, and SKT signal were normalized separately using Equation (1):(1)$$\tilde{x}=\frac{x-\mu }{\sigma }$$

#### 2.2.2. Ectopic and Missing Value Processing

For ectopic values, ectopic value detection was performed first. The percentage detection method, which defines data points with more than 20% variation from the previous data point as ectopic, was adopted. Time windows (30 s) with more than 20% ectopic or missing values were removed [26,27]. The remaining ectopic and missing values were replaced with the mean value of the 11 adjacent data points centered on the missing or ectopic value using Equation (2):(2)$$x{}^{\prime }\left(n\right)=mean\left(x\left(n+m\right)\right),\;\mathrm{where}\;\left|m\right|\leq \frac{w-1}{2}$$

#### 2.2.3. Downsampling and Filtering

Since the sampling rate of physiological signals is usually much higher than what is needed and the sampling rates vary [28], downsampling is needed. Before downsampling, heart rate variability (HRV) features and frequency domain features were extracted first. Then, to ensure the data consistency in timing and the ability to represent the pilot’s behavior [29], all physiological signals were downsampled to 2 Hz based on the minimum time of flight maneuvers (5 s for subject 2′s roll maneuver).

In addition, the physiological data were filtered to reduce interference from motion artifacts, human noise, and other factors. Filtering mainly includes noise reduction, high pass, band resistance, and low pass. Wavelet noise reduction was employed to remove the baseline noise and drift signal from the signal. Gaussian filters, on the other hand, transform the data by building a mathematical model, smoothing it, and reducing the effect of noise [30,31,32,33,34]. The detailed data processing methods are shown in Table 2.

### 2.3. Data Conversion

Data conversion means transforming the data into a suitable form for data mining through aggregation. To transform the processed data into features that can be input into the classifier and to make the features accurately describe the data, the physiological data were analyzed in the time and frequency domains. To ensure the accuracy and continuity of time- and frequency-domain data analysis, we selected 30 s as the time window and 10 s as the step to obtain the continuous time- and frequency-domain indexes.

#### 2.3.1. Time-Domain Features

Time-domain information is a depiction of the signal waveform with time as the variable. Time-domain features include dimensional characteristic parameters, as well as dimensionless characteristic parameters [35]. In this paper, the main parameters used were the dimensional characteristics, including the indicators associated with heart rate variability, the mean, the standard deviation, and the RMS. The expressions are shown in Table 3.

#### 2.3.2. Frequency-Domain Features

Frequency-domain analysis observes signal characteristics by frequency. The analysis in the time domain is more intuitive, while the representation in the frequency domain is more concise [36]. The frequency-domain feature parameters used in this study included HRV frequency-domain analysis, EMG signal frequency-domain analysis, and RESP signal frequency-domain analysis. The expressions are shown in Table 4.

#### 2.3.3. Multi-Modal Features Conversion

The pilot’s behaviors are reflected in physiological and biological changes, including changes in heart beat, muscle activity, respiration, reflexes, etc. [37]. Therefore, these features were extracted from ECG, GSR, EMG, RESP, and SKT. Since physiological measurements change as the pilot’s behavior changes, statistical features were extracted to describe the variation in the measurements. As such, the variations of the time-domain features and frequency-domain features commonly used in ECG, GSR, EMG, RESP, and SKT measurements were analyzed to determine the most relevant features. The mathematical representation of these features extracted from the multi-dimensional signal measurements per time window is shown in Table 3 and Table 4. Then, all features were fused according to the time series. A total of 28 features, as shown in Table 5, were derived at each time point.

#### 2.3.4. Correlation Analysis

To ensure the independence of the features, the degree of correlation between the features must be analyzed. Pearson correlation coefficient *r* reflects the degree of linear correlation between two variables *x* and *y*. The value of r is between −1 and 1, and the larger the absolute value, the stronger the correlation [38,39]. The formula for the Pearson correlation coefficient r is shown in Equation (3):(3)$$r_{x,y}=\frac{cov\left(x,y\right)}{\sigma_{x}\sigma_{y}}=\frac{E\left(\left(x-\mu_{x}\right)\left(y-\mu_{y}\right)\right)}{\sigma_{x}\sigma_{y}}=\frac{E\left(xy\right)-E\left(x\right)E\left(y\right)}{\sqrt{E\left(x^{2}\right)-E^{2}\left(x\right)}\sqrt{E\left(y^{2}\right)-E^{2}\left(y\right)}}$$

Figure 3 is the correlation heatmap of the features. The different colors in the heatmap correspond to the correlation coefficient; the darker the color, the greater the correlation between the corresponding two features. The model tends to be influenced and to output wrong results if there is high correlation between features [40].

According to the Pearson correlation coefficients among the features, it can be seen that most of the features had low or no correlation, and a moderate correlation existed among SDNN, RMSSD, SDSD, ULF, VLF, and LF. High correlations were observed in HR and NN, SDSD and RMSSD, SC and SC mean, and EMG mean and iEMG. Features with high correlations were then removed in the Classification Improvement part of the procedure.

### 2.4. Modeling

The features selected in Data Conversion part of the procedure were used as input for the classifier models. The model suitable for this study was selected based on the advantages and disadvantages of machine learning models. Several commonly used classifiers were trained on the data set, and their 10-fold cross-validation results are shown in Table 6. Candidate classifiers were picked by analyzing the validation results and reviewing previous studies [17,18,19,41,42,43]. Finally, 4 classifiers were selected due to their high accuracy and low MSE, including the Extra Tree Classifier (ETC), Decision Tree Classifier (DTC), Gradient Boosting Classifier (GBC), and XGBoost (XGBC). Extra Tree is a modification of Random Forest (RF); its principle is the same as RF, but its generalization ability is stronger than RF. DTC is an algorithm that divides the input space into different regions. Compared with other machine learning classification algorithms, DTC is relatively simple and efficient in data processing, which makes it suitable for real-time classification. GBC is an algorithm for regression and classification problems that integrates weak predictive models to produce a strong predictive model. XGBC is its enhanced version. The residuals of prediction are reduced so that the effect can be improved.

The goal of the modeling was to find the best classifier suitable for the study. To integrate the predictions of these classifiers and, thus, improve the prediction performance, a model was proposed that used a decision layer fusion technique. The advantage of decision level fusion is that the errors of the fusion model come from different classifiers, and the errors are often unrelated and independent of each other, which does not cause further accumulation of errors [44]. As such, we created our proposed model by assigning different weights to these classifiers and feeding their predictions to an integrated model, which is also known as ensemble learning. The voting algorithm is one of the simplest, most popular, and effective combiner schemes for ensemble learning [45,46]. It fuses the results from various learning algorithms to achieve knowledge discovery and better predictive performance [47,48,49]. Generally, there are two types of voting algorithm, majority voting (MV) and weighted voting (WV) [50]. Past studies showed the effectiveness of ensemble learning over the learning of a single learner [51,52]. After careful investigation, we proposed a model based on a weighted version of the simple majority voting, where each classifier contributes to the final output according to a reasonable weight ω. ω is calculated by the base classifiers along with the related, estimated probability distributions. It is also known as the confidence level (CL), indicating the degree of support for the prediction. The framework and scheme of the model is shown in Figure 4 and Algorithm 1.
**Algorithm 1.** Weighted voting scheme.**Input**:$C_{i}$: Classifier$L_{j}$: Labels of Data Setm: Ensemble Sizen: the Number of Labels**Output**:the predicted class $y_{j}$ from a single classifier $C_{i}$the predicted class y***for** i = 1: m   **for** j = 1: n      **compute** $p_{C_{i}L_{j}}$, the probability assigned by $C_{i}$ to class $L_{j}$   μ = ${\arg\;\max}_{L=1,\ldots ,r}$ $p_{C_{i}L_{j}}$   $y_{C_{i}}$ = $y_{\mu }$**for** j = 1: n   $y\left(L_{j}\right)$ = { i = 1,…,m: $y_{C_{i}}$ == $y_{L_{j}}$}   **if** $y\left(L_{j}\right)$ == ∅      $g_{L_{j}}$ = 0   **else**      **for** i in $y\left(L_{j}\right)$ do         $\omega_{C_{i}}$ = $\max_{j=1,\ldots ,n}$ $p_{C_{i}L_{j}}$
      $g_{L_{j}}$= $\overset{m}{\underset{i=1}{\sum }}\omega_{C_{i}}$,μ = ${\arg\;\max}_{L=1,\ldots ,r}$ $g_{L_{j}}$y* = $y_{\mu }$   **return** y*

## 3. Results and Discussion

### 3.1. Data Measures

Table 7 shows the results of the subjective difficulty assessment provided by the participants (assessed while watching a video playback directly following the flight).

Time-domain measurements, such as NN, SDNN, RMSSD, pNN50, pNN20, the mean of RESP, the standard deviation of RESP, the mean of GSR, the standard deviation of GSR, the mean of EMG, the standard deviation of EMG, the RMS of EMG, and iEMG, and frequency-domain measurements, such as ULF, VLF, LF, HF, LF/HF, the frequency of RESP, the EMG median frequency, and the EMG mean power frequency, as well as raw data, such as ECG, HR, RESP, SC, EMG, and SKT, were recorded according to pilot behavior and plotted as a trend (Figure 5).

To objectively analyze the relationship between these physiological parameters and different behaviors, we analyzed them using the Kendall correlation analysis [53], as shown in Figure 6. It turned out that the ECG value, HR, SDNN, SDSD, pNN50, pNN20, VLF, LF, HF, RESP value, REAP mean, RESP standard deviation, SC standard deviation, EMG value, RMS, iEMG, EMG mid frequency, EMG mean frequency, and SKT value had no evident correlations with behavior; LF/HF, SCL, the EMG mean, and EMG standard deviation had moderate correlations with behavior; and NN, ULF, RESP frequency, SCL, and the SCL mean had high correlations with behavior. Parameters with a dark-green background were negatively correlated with the degree of difficulty, and parameters with a light-green background were positively correlated with the degree of difficulty. The trends in physiological parameters were consistent with previous studies [54,55,56,57,58]. Nevertheless, the results showed slight differences in ULF compared to some studies [9,59,60,61,62] which believed that ULF cannot reflect a human’s workload level. After further literature research and analysis, our results were found to be justified. The pilot is in a complex state when performing flight maneuvers, inducing complicated physiological responses associated with cognition, fatigue, stress, concentration, or distraction, etc. So, studies which only take workload into consideration are neither comprehensive nor convincing with regard to difficulty rating. As a result, despite ULF being unable to reflect a human’s workload level, it could possibly be used in difficulty level recognition.

The objective analysis of the correlation between physiological indicators and behaviors helped to select key indicators for research. For example, if the condition was too limited to collect multiple physiological features, the features with higher correlation were selected for analysis. Additionally, this is the visual explanation of the feature importance analysis.

### 3.2. Classification Model Performance

Figure 7 and Table 8 depict the 10-fold cross-validation (10-fold CV) results and leave-one-person-out (LOO) cross-validation results of various classifiers. The principles of LOO CV can be found in Appendix A [63]. In this study, the proposed model obtained a higher accuracy and generalization than other machine learning classification models. Although the mean accuracy of the proposed model was not much improved compared to the base model, the results of cross-validation show that it had the best stability. In addition, the robustness and generalization of the model was improved.

### 3.3. Classification Improvement

The importance ranking of various classifiers for features was found by feature importance selection, as shown in Figure 8.

Removing irrelevant features can help to ease the learning task, make the model simple and reduce the computational complexity. Based on the importance ranking, we chose to keep NN, SDNN, pNN50, pNN20, VLF, LF, HF, RESP, RESP mean, RESP standard deviation, SC, SC mean, SC standard deviation, EMG mean, EMG standard deviation, EMG RMS, iEMG, EMG median frequency, and EMG mean power frequency as features. After the features were selected and re-input into the model, the classification performance was as shown in Figure 9 and Table 9. It was found that, after feature selection, the performance of all models improved.

According to the classification report, ‘takeoff’ achieved the highest accuracy, probably because takeoff is at the beginning of the flight when the pilot’s attention is most focused. To obtain more reliable data, in future work, we plan to perform the takeoff operation several times in one trial to avoid the effect of attention on the results. Confusion tended to occur between ‘level flight’, ‘roll’, and ‘turn and hover’, probably due to the similar difficulty of these behaviors and the fact that stalls rarely occur during these maneuvers. To our surprise, the ‘somersault’ was sometimes confused with ‘level flight’. After discussion with the pilots, we thought that it might have been because there was a period of level flight before and after the somersault. So, the definition of the flight behavior still needed further clarification. ‘Stall’ achieved the lowest accuracy. We believe this is due to the minimum number of sample points in the ‘stall’ state, which led to an insufficient amount of data and, thus, affected the results. Therefore, expanding the data set is necessary to train a better model.

Another finding is that the classification model trained on a data set using 10-fold CV does not perform best in leave-one-person-out cross-validation. The reason for the variation might be attributed to individual differences. Hence, how to select the most suitable classifier based on the results of 10-fold cross-validation and LOO cross-validation and how to build an adaptive model to reduce individual variability will be the focus of subsequent research.

## 4. Conclusions

This study provided a comparison of the physiological responses of 14 pilots performing flight missions with different flight behaviors, thus, demonstrating the physiological indicators influenced by the pilot’s behavioral status and their association with flight difficulty. It was found that ECG value, HR, SDSD, pNN50, pNN20, VLF, LF, HF, RESP value, REAP mean, RESP standard deviation, SC standard deviation, EMG value, RMS, iEMG, EMG mid frequency, EMG mean frequency, and SKT had no evident relations with pilots’ behaviors. RESP frequency, SCL, and SCL mean rose with an increasing subjective difficulty rating. On the contrary, NN, ULF, LF/HF, EMG mean, and EMG standard deviation decreased with increasing subjective difficulty rating. This provided a clue for selecting key indicators for research, especially as the situation was too limited to collect multiple physiological features. The comparison also visually explained the selection of feature importance, which helped to reduce redundancy, avoid overfitting, and improve the real-time detection capability of the model.

Furthermore, a model that recognizes pilots’ behavior based on multi-modal fusion technology (MFT) using physiological characteristics was proposed. Pilot multi-modal physiological parameters and various machine learning classifiers were used to detect pilot behavior. Several machine learning models were employed to recognize the pilot’s behaviors. By analyzing and comparing the accuracy, precision, F1 score, and MSE of different models, it was found that the pilot-state-recognition model could be improved in many ways, including feature layer fusion, decision layer fusion, and feature filtering. The experimental results verified the superiority of the proposed model in recognizing flight behavior. The accuracy of the proposed model reached 98.15%, proving that MFT is promising for pilot state recognition.

Due to the limited data set used in this study, the accuracy of the classification network could not be fully verified. In future work, it will be necessary to acquire more data for training and testing to verify the model more precisely. Building an adaptive model to reduce individual variation is also the core of future work. In addition, we plan to apply the model in practice to detect the effects of flight assist systems on the physiological state of pilots when they perform various behaviors. Furthermore, the generalization of the model will be improved to make it applicable to more scenarios.

## Figures and Tables

**Figure 1 biosensors-12-00404-f001:**
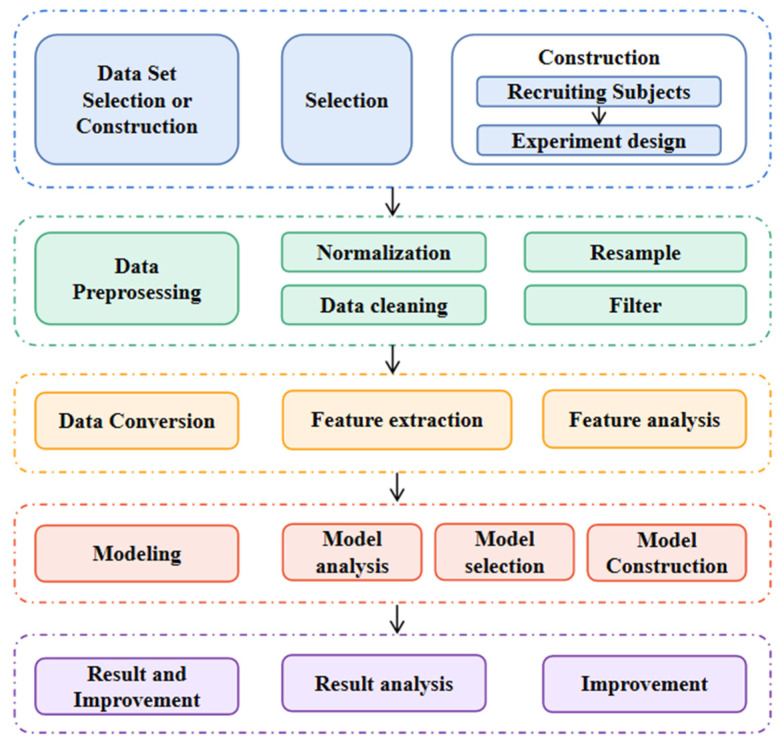
The process of data mining.

**Figure 2 biosensors-12-00404-f002:**
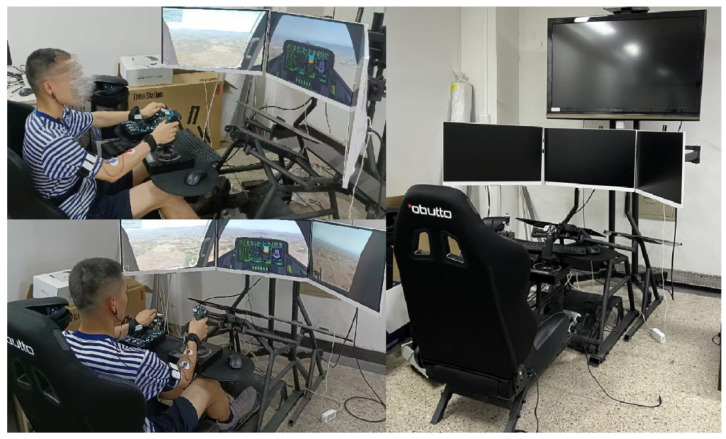
Illustration of the experimental platform.

**Figure 3 biosensors-12-00404-f003:**
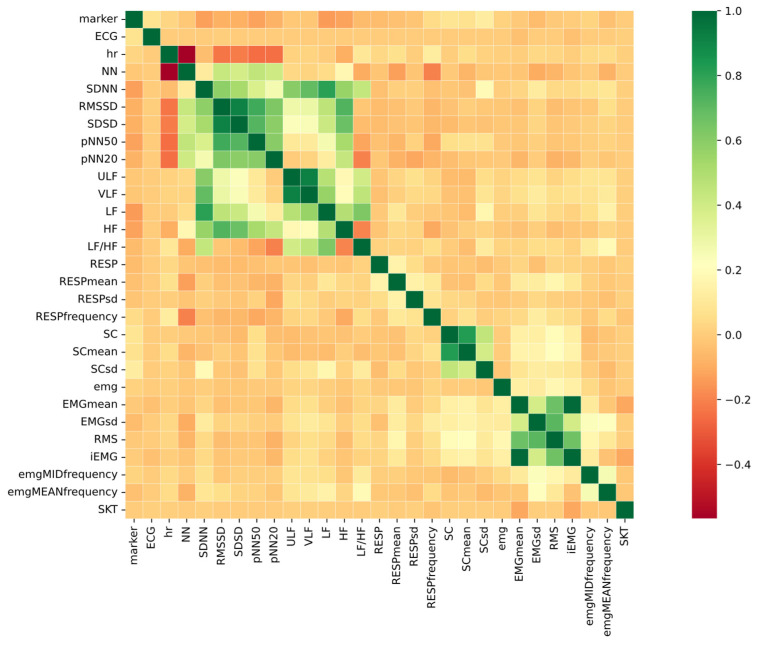
Pearson correlation analysis.

**Figure 4 biosensors-12-00404-f004:**
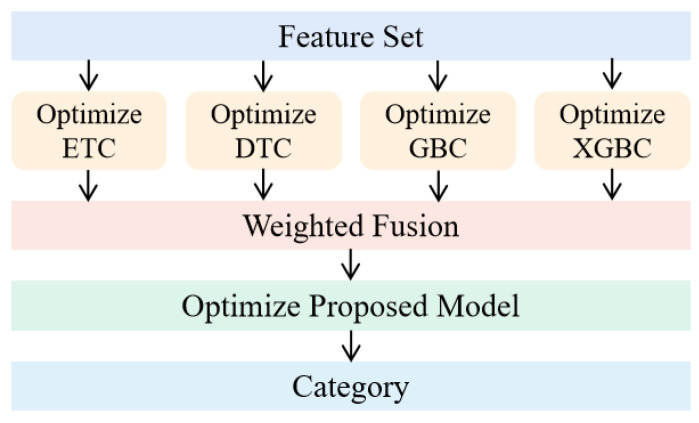
The framework of proposed model.

**Figure 5 biosensors-12-00404-f005:**
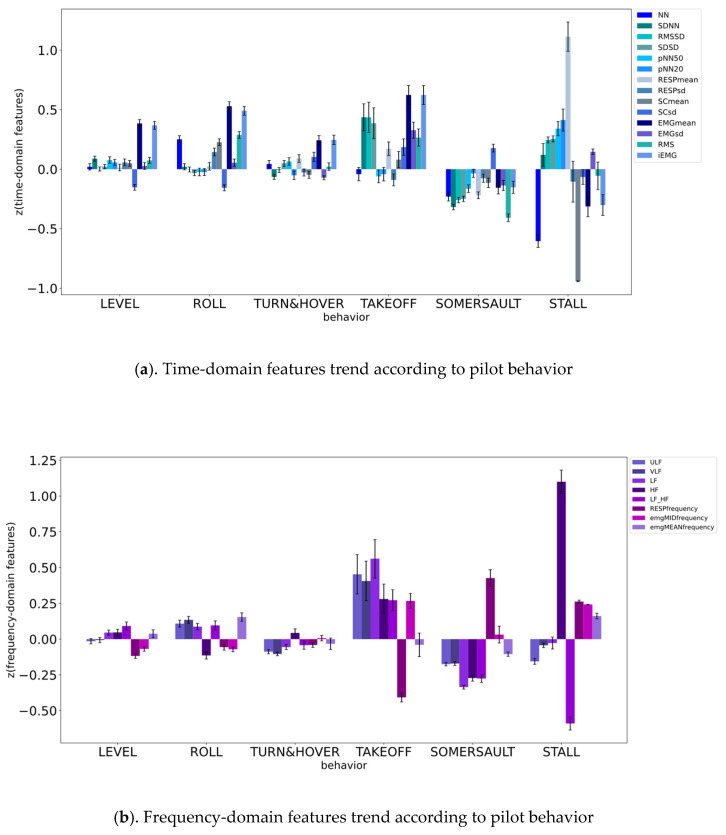
Data trend according to pilot behavior.

**Figure 6 biosensors-12-00404-f006:**
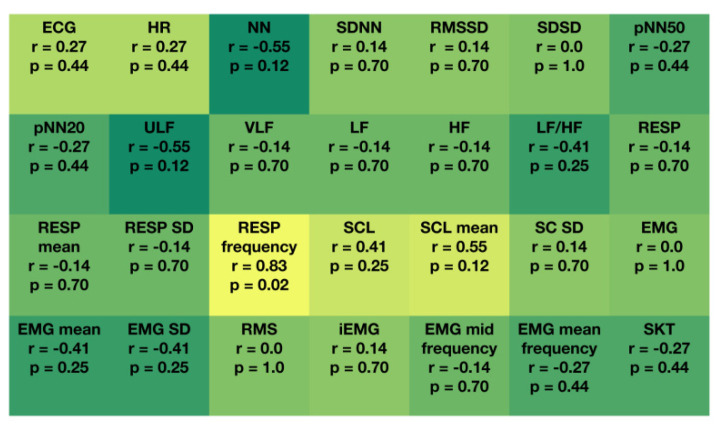
Kendall correlation analysis.

**Figure 7 biosensors-12-00404-f007:**
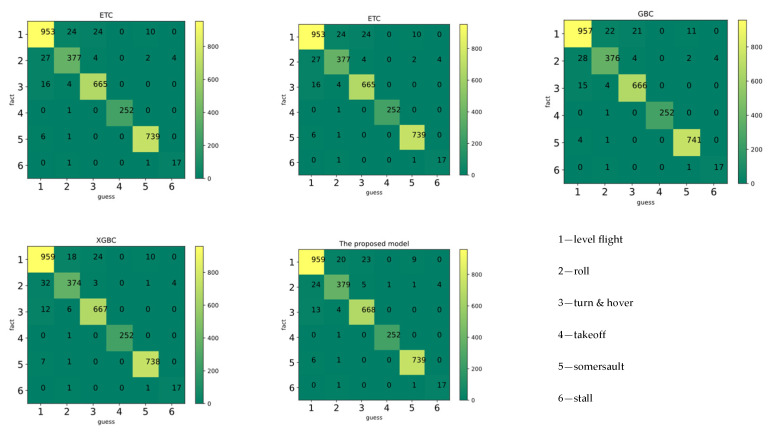
The confusion matrix of classifiers.

**Figure 8 biosensors-12-00404-f008:**
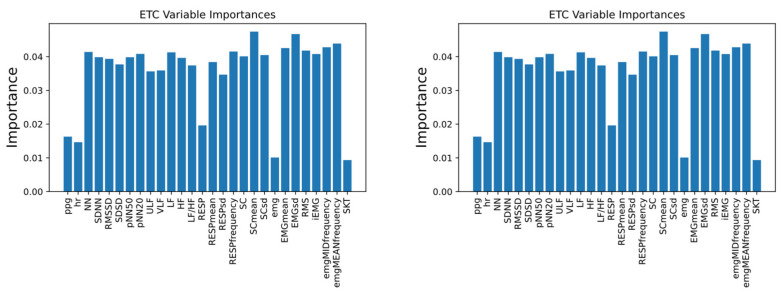
The importance ranking of the four different classifiers.

**Figure 9 biosensors-12-00404-f009:**
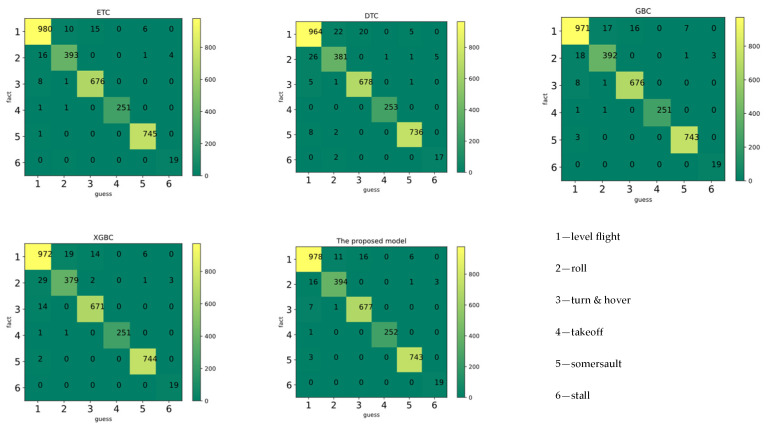
The confusion matrix of classifiers after improvement.

**Table 1 biosensors-12-00404-t001:** Data pre-processing methods.

Category	Aim	Methods
Data cleaning	Handling of anomalies in data values	Missing value processing (abandon/replacement)
		Ectopic values processing
		Outlier and noise handling
Data integration	Increase sample data size	Combining multiple data sets into a single data set
Data standardization	Scales the sample values to a specified range	Discretization
		Dualization
		Normalization (min–max, z-score)
		Function transformation

**Table 2 biosensors-12-00404-t002:** Detailed data processing.

	ECG	GSR	EMG	RESP	SKT
Noise reduction	Wavelet	Gaussian	Wavelet	Wavelet	Sliding average
High pass	1 Hz	/	5 Hz	/	5 Hz
Band stop	50 Hz	50 Hz	50 Hz	50 Hz	50 Hz
Low pass	40 Hz	5 Hz	500 Hz	20 Hz	200 Hz

**Table 3 biosensors-12-00404-t003:** Time-domain features mathematical representation.

Parameters	Description
Mean	$\bar{x}=\frac{i}{N_{s}}\overset{N_{s}}{\underset{i=1}{\sum }}x\left(i\right)$
Standard Deviation	$F_{s}=(\frac{1}{N_{s}}\sqrt{\overset{N_{s}}{\underset{i=1}{\sum }}{\left(x\left(i\right)-\bar{x}\right)}^{2}}$
Root Mean Square (RMS)	$F_{s}=\sqrt{\frac{1}{N_{s}}\overset{N_{s}}{\underset{i=1}{\sum }}{\left(x\left(i\right)\right)}^{2}}$

**Table 4 biosensors-12-00404-t004:** Frequency-domain features mathematical representation.

Parameters	Description
Power	Power in the frequency band
Median Frequency	$\int_{0}^{MF}P\left(\omega \right)d\omega =\int_{MF}^{\infty }P\left(\omega \right)d\omega =\frac{1}{2}\int_{0}^{\infty }P\left(\omega \right)d\omega$
Mean Power Frequency	$MPF=\frac{\int_{0}^{\infty }\omega P\left(\omega \right)d\omega }{\int_{0}^{\infty }P\left(\omega \right)d\omega }$

**Table 5 biosensors-12-00404-t005:** Multi-modal features.

ECG		GSR	EMG	RESP	SKT
ECG value	HR value	SC value	EMG value	RESP value	SKT value
SDSD	NN	mean	standard deviation	standard deviation	
SDNN	RMSSD	standard deviation	RMS	power	
pNN50	pNN20		Integral EMG	mean	
VLF	ULF		median frequency		
LF	HF		mean power frequency		
LF/HF			mean		

The full details of abbreviations can be found in Abbreviations.

**Table 6 biosensors-12-00404-t006:** The performance of several classification models.

Model	Mean Accuracy	Lowest Accuracy	MSE
Logistic Regression	0.430	0.417	4.0109
Naive Byes	0.362	0.339	4.1813
AdaBoost	0.373	0.355	4.0822
SVM	0.441	0.432	3.9632
K-Nearest Neighbor	0.952	0.947	0.2989
ETC	0.965	0.962	0.1765
DTC	0.964	0.960	0.1733
GBC	0.968	0.965	0.1755
XGBC	0.967	0.962	0.1847

**Table 7 biosensors-12-00404-t007:** Difficulty level rating according to subjective ratings.

	Stall	Somersault	Takeoff	Turn and Hover	Level Flight	Roll
Subject 1	10	8	6	6	4	3
Subject 2	7	6	8	5	3	4
Subject 3	8	9	5	6	4	4
Subject 4	9	8	3	5	3	3
Subject 5	8	5	2	3	2	2
Subject 6	8	4	1	4	1	3
Subject 7	6	8	1	5	2	2
Subject 8	3	7	2	3	3	6
Subject 9	8	9	5	6	5	5
Subject 10	7	8	3	6	4	4
Subject 11	8	7	3	5	4	3
Subject 12	5	7	2	6	5	2
Subject 13	7	9	3	7	5	3
Subject 14	8	8	4	5	4	3
Mean	7.29	7.36	3.43	5.14	3.50	3.36

**Table 8 biosensors-12-00404-t008:** The classification report of classifiers.

	**ETC**					
	level	roll	turn and hover	takeoff	somersault	stall
**precision**	0.96	0.91	0.97	1.00	0.99	0.97
**recall**	0.95	0.92	0.98	1.00	0.99	1.00
**F1**	0.96	0.91	0.97	1.00	0.99	0.98
**average accuracy for 10-fold CV**			0.9652	**MSE for 10-fold CV**		0.1765
**average accuracy for LOO CV**			0.7817	**MSE for LOO CV**		1.1199
	**DTC**					
	level	roll	turn and hover	takeoff	somersault	stall
**precision**	0.96	0.91	0.97	1.00	0.99	0.97
**recall**	0.95	0.92	0.97	1.00	0.99	1.00
**F1**	0.96	0.91	0.97	1.00	0.99	0.98
**average accuracy for 10-fold CV**			0.9642	**MSE for 10-fold CV**		0.1733
**average accuracy for LOO CV**			0.7062	**MSE for LOO CV**		1.5760
	**GBC**					
	level	roll	turn and hover	takeoff	somersault	stall
**precision**	0.96	0.91	0.97	1.00	0.99	0.97
**recall**	0.95	0.93	0.98	1.00	0.99	0.93
**F1**	0.96	0.92	0.98	1.00	0.99	0.95
**average accuracy for 10-fold CV**			0.9677	**MSE for 10-fold CV**		0.1755
**average accuracy for LOOCV**			0.7064	**MSE for LOO CV**		1.4856
	**XGBC**					
	level	roll	turn and hover	takeoff	somersault	stall
**precision**	0.96	0.91	0.97	1.00	0.99	0.97
**recall**	0.96	0.92	0.98	1.00	0.99	1.00
**F1**	0.96	0.92	0.98	1.00	0.99	0.98
**average accuracy for 10-fold CV**			0.9674	**MSE for 10-fold CV**		0.1847
**average accuracy for LOO CV**			0.7473	**MSE for LOO CV**		1.4894
	**Proposed Model**					
	level	roll	turn and hover	takeoff	somersault	stall
**precision**	0.96	0.93	0.97	1.00	0.99	0.97
**recall**	0.96	0.92	0.98	1.00	0.99	1.00
**F1**	0.96	0.93	0.98	1.00	0.99	0.98
**average accuracy for 10-fold CV**			0.9693	**MSE for 10-fold CV**		0.1693
**average accuracy for LOO CV**			0.8094	**MSE for LOO CV**		1.0606

**Table 9 biosensors-12-00404-t009:** The classification report of classifiers after improvement.

	**ETC**					
	level	roll	turn and hover	takeoff	somersault	stall
**precision**	0.97	0.95	0.99	1.00	1.00	1.00
**recall**	0.97	0.96	0.99	1.00	0.99	0.91
**F1**	0.97	0.96	0.99	1.00	0.99	0.95
**average accuracy for 10-fold CV**			0.9792	**MSE for 10-fold CV**		0.1093
**average accuracy for LOO CV**			0.7889	**MSE for LOO CV**		1.1933
	**DTC**					
	level	roll	turn and hover	takeoff	somersault	stall
**precision**	0.96	0.92	0.99	1.00	0.99	0.97
**recall**	0.96	0.94	0.98	1.00	0.99	0.88
**F1**	0.96	0.93	0.98	1.00	0.99	0.92
**average accuracy for 10-fold CV**			0.9728	**MSE for 10-fold CV**		0.1579
**average accuracy for LOO CV**			0.7301	**MSE for LOO CV**		1.3759
	**GBC**					
	level	roll	turn and hover	takeoff	somersault	stall
**precision**	0.98	0.93	0.99	1.00	0.99	0.97
**recall**	0.97	0.96	0.98	1.00	0.99	0.90
**F1**	0.97	0.94	0.99	1.00	0.99	0.93
**average accuracy for 10-fold CV**			0.9726	**MSE for 10-fold CV**		0.1266
**average accuracy for LOO CV**			0.7306	**MSE for LOO CV**		1.2341
	**XGBC**					
	level	roll	turn and hover	takeoff	somersault	stall
**precision**	0.96	0.93	0.98	1.00	1.00	1.00
**recall**	0.97	0.94	0.98	1.00	0.99	0.94
**F1**	0.97	0.94	0.98	1.00	0.99	0.97
**average accuracy for 10-fold CV**			0.9741	**MSE for 10-fold CV**		0.1151
**average accuracy for LOO CV**			0.7697	**MSE for LOO CV**		1.3256
	**Proposed Model**					
	level	roll	turn and hover	takeoff	somersault	stall
**precision**	0.98	0.95	0.99	1.00	1.00	0.97
**recall**	0.98	0.96	0.98	1.00	1.00	0.93
**F1**	0.98	0.95	0.99	1.00	1.00	0.95
**average accuracy for 10-fold CV**			0.9815	**MSE for 10-fold CV**		0.1026
**average accuracy for LOO CV**			0.8273	**MSE for LOO CV**		0.9601

## Data Availability

The data presented in this study are available on request from the corresponding author. The data are not publicly available due to privacy.

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
