# Peer review of "Pilot Behavior Recognition Based on Multi-Modality Fusion Technology Using Physiological Characteristics"

_biosensors, 2022, doi:10.3390/bios12060404_

Round 1

Reviewer 1 Report

The subject addressed in this article is interesting and has many applications in aeronautical field (pilot training, interaction between pilots and plane control equipment, etc.)

After reading and analyzing the article, I can make the following observations:

- In introduction the authors state that "the research on multi-modal physiological signals for human state recognition is still at the very beginning" that is not true - at least in the emotion recognition field there are many studies that involve combinations of physiological signals:

----------

Šalkevicius, J.; Damaševiˇcius, R.; Maskeliunas, R.; LaukienË™ e, I. Anxiety Level Recognition for Virtual Reality Therapy System Using Physiological Signals. Electronics 2019, 8, 1039.

Petrescu, L.; Petrescu, C.; MitruÈ›, O.; Moise, G.; Moldoveanu, A.; Moldoveanu, F.; Leordeanu, M. Integrating Biosignals Measurement in Virtual Reality Environments for Anxiety Detection. Sensors 2020, 20, 7088. https://doi.org/10.3390/s20247088

Chen, L.-L.; Zhao, Y.; Ye, P.-F.; Zhang, J.; Zou, J.-Z. Detecting driving stress in physiological signals based on multimodal feature analysis and kernel classifiers. Expert Syst. Appl. 2017, 85, 279–291.

Granato, M.; Gadia, D.; Maggiorini, D.; Ripamonti, L.A. Feature Extraction and Selection for Real-Time Emotion Recognition in Video Games Players. In Proceedings of the 2018 14th International Conference on Signal-Image Technology & Internet-Based Systems (SITIS), Las Palmas de Gran Canaria, Spain, 26–29 November 2018; pp. 717–724.

---------

- Simulation environment is not very realistic. It does not generates the hight G-forces present during real flights (especially for military aiircrafts). The authors may comment the effect of these forces on human cardiac and respiratory systems and how the pilot's behaviour recognition will be affected in the presence of these forces.

- The authors may state the aim of the study more clear. In the current form, the reader can undersand that the aim is to estimate the task difficulty from physiological data not identification of the pilot action (takeoff, level flight, hover and turn).

- In many studies, normalization is performed relative to baseline values (parameter's values during resting/relaxed conditions). The method proposed in the article leads to physiological signals havind zero mean and unity standard deviation. It would be useful to add a short argumentation for selecting this method.

- How large is the time window used for ectopic value?

- Downsampling ECG signal to 2 samples/sec will make impossible to observe ECG signal features and extract heart rate information. Some supplementary informations are needed related to downsampling process, maybe some signal features are extracted before downsampling.

- Data conversion term seems to be confusing. A more apropriate term could be for example signal's time segmentation.

- In table 5, VLF, ULF, LF, HF, LF/HF features needs to be detailed by specifying the frequency thresholds used.

- How many experiments are performed by each subject. What is the total size of the dataset used for trainig and testing?

- Figure 7 is not very clear (some numbers can not be read).

In conclusion, I recommend to reconsider the article after major revision.

Author Response

Dear reviews and editor,

Thank you for taking your time to review this manuscript. I really appreciate all your comments and suggestions! Please see my reply to the review report in the attachment.

Thanks again!

Reviewer 2 Report

It is interesting to tackle with a specific behavior recognition task, i.e., pilot behavior. However, a number of issues were found:

1)     Which is the objective of the work, i.e., pilot behavior recognition or difficulty level estimation? Most of the content describes about behavior recognition. But, Table 6 suddenly appears, and a sentence “The experimental results verify the superiority of the proposed model in recognizing and predicting the difficulty of flight behavior.” Appeared in Section 4 (line 366-367).

2)     What is “the proposed model”? The answer seems to the ensemble classification method; however, it is not described concretely, just with a pseudo code. Unfortunately, the code does not describe the idea enough. It is not clear how the weights are determined (what is “reasonable weight” in line 272?), what the optimization of proposed model mean, and how the ensemble method differs from existing weighted ensemble methods. Actually, it is not clear what is included in the “proposed model”; are the choice of the individual classifier also the contribution. If yes, various combinations of the candidate classifiers should be evaluated. From my point of view, it is curious to use DTC and other tree-based model, because DTC has already included in the ensemble tree-model.

3)     What does the “classification network” mean? Does the proposed method form a network?

4)     How many pilots participated in the data collection? In line 94 and Table 6, it is 14, but it is three in line 354.

5)     The explanations for each figure in Figures 5 and 8 should be added. Also, the characters in the figures are too small to read.

6)     The proposed method shows the highest classification performance; however, the difference seems very small. The significance should be tested using appropriate statistical test method.

7)     The evaluation was performed 80% training and 20% testing scheme (line 244). Does it mean a 5-fold cross validation that repeats the combination of 80%training-20%test five times, or just one time training and testing with specific samples? If it is not cross validation, the resultant performance can be a product of chance and unreliable. Regarding the evaluation method, it is not clear robustness of the proposed method on the individual difference. Please do a so-called leave-one-person-out cross validation to understand the individual difference and the pessimistic, i.e., lower limit of, performance in addition to k-fold cross validation.

8)     What are the key evaluation metrics? In line 93, they seem to be accuracy, precision, and recall; however, in Tables 7 and 8, F1-score also appeared.

9)     In section 4 (conclusion) (lines 355-361), the findings on the correlation of features from various physiological sensors are described. What is the contribution of the findings? How do they contribute to the final goal of recognizing the pilot behavior?

10) “Table 2 and Table 3” in line 219 should be “Table 3 and Table 4”?

11) How many features are defined actually? In Table, it seems to be 27, but in line 220 it is 28.

So, I consider that the manuscript is not yet ready for publication in the journal.

Author Response

Dear reviews and editor,

Thank you for reviewing this manuscript. I really appreciate your comments and suggestions! Please find my itemized responses in the attachment.

Thanks again!

Round 2

Reviewer 1 Report

After reading the revised article and author's comments, I noticed that all observations were properly addressed or explained.

One minor observation is related to references for figures and tables that appear as "Error! Reference source not found." 

In conclusion, I recommend to accept the article in present form.

Reviewer 2 Report

I have now read the reply from the author and confirmed that they addressed my concern. So, I agree with acceptance.